# A multichromatic colorimetric detection method for *Vibrio parahaemolyticus* based on Fe₃O₄-Zn-Mn nanoenzyme and dual substrates

Wenteng Qiao,[1] Luliang Wang,[1,2] Kun Yang,[1] Yushen Liu,[1,2,3] Quanwen Liu,[1] Feng Yin[4]

**ABSTRACT**    A successful development of a dual-substrate colorimetric system for rapid and multicolorimetric semi-quantification of *Vibrio parahaemolyticus (V. parahaemolyticus)* has been achieved. The enzymatic activity of Fe₃O₄-Zn-MnO₂-aptamer (Fe₃O₄-Zn-Mn-Apt) catalysts can induce the oxidation of 3,3′,5,5′-tetramethylbenzidine (TMB) to form oxidized TMB (TMB⁺ with a bluish green color) and ortho-phenylenediamine (OPD) to form oxidized OPD (with yellow color), resulting in a three-color comparison due to the complementary nature of bluish green and yellow. The absorbance value of TMB⁺ at 652 nm is used for the determination of *V. parahaemolyticus*. After optimizing the reaction conditions, the developed dual-substrate colorimetric method exhibits high sensitivity for *V. parahaemolyticus* detection (limit of detection of 1.12 cfu mL⁻¹) and a linear range of $0–1 \times 10^4$ cfu mL⁻¹ ($R^2$ = 0.9934). Additionally, as the concentration of *V. parahaemolyticus* increases, the reaction solution changes from bluish green to green and then to yellow, enabling semi-quantitative detection by visual observation with a visual detection limit of 10 cfu mL⁻¹. Furthermore, the developed dual-substrate colorimetric method demonstrates excellent selectivity for *V. parahaemolyticus* detection and satisfactory recovery rates when applied to the determination of *V. parahaemolyticus* in food samples.

**IMPORTANCE**   The Fe₃O₄-Zn-Mn nanomimetic enzyme demonstrates significant importance in dual-substrate colorimetric detection for *V. parahaemolyticus*, owing to its enhanced sensitivity, selectivity, and rapid detection capabilities. Additionally, it offers cost-effectiveness, portability, and the potential for multiplex detection. This innovative approach holds promise for improving the monitoring and control of *V. parahaemolyticus* infections, thereby contributing to advancements in public health and food safety.

**KEYWORDS**   Fe₃O₄-Zn-Mn nanoenzyme, dual substrates, multicolorimetric, *Vibrio parahaemolyticus*

*V*ibrio parahaemolyticus (*V. parahaemolyticus*), a prevalent foodborne pathogen, thrives abundantly in both marine and freshwater environments (1, 2). It serves as the primary causative agent of *VP* infection, which manifests as severe gastrointestinal distress characterized by intense abdominal pain, diarrhea, nausea, and vomiting. In certain cases, the infection can lead to grave complications such as sepsis and septic shock (3, 4). The transmission of *V. parahaemolyticus* primarily occurs through the consumption of contaminated seafood. These edibles are susceptible to bacterial contamination during collection, processing, and storage, thus becoming the primary route of infection (5). Additionally, *V. parahaemolyticus* can also spread through contact with contaminated water sources or direct exposure to infected individuals' feces. With the growth of global seafood trade and increased consumption of seafood, the incidence

Address correspondence to Yushen Liu, yushenlys@163.com, or Feng Yin, 1282128466@qq.com.

Wenteng Qiao and Luliang Wang contributed equally to this article. Author order was determined by division of work.

The authors declare no conflict of interest.

See the funding table on p. 10.

of *V. parahaemolyticus* infection is on the rise (6). Particularly in warm climatic conditions, bacterial proliferation accelerates, exacerbating food safety concerns. Consequently, the detection and control of *VP* have become increasingly crucial.

Currently, commonly employed methods for *V. parahaemolyticus* detection encompass traditional culture-based techniques and molecular biology methods (7). Traditional culture-based methods involve inoculating samples (such as seafood and water sources) onto specific culture media and employing bacterial growth characteristics for screening and identification (8). This method necessitates a certain amount of time for bacterial cultivation, typically requiring 24–48 hours to yield results (9). Although traditional culture-based methods are widely utilized, they possess certain limitations, including the need for extended time and stringent bacterial cultivation conditions. Common molecular biology methods encompass polymerase chain reaction, real-time fluorescence polymerase chain reaction, and gene sequencing, among others (10). These techniques enable rapid and accurate detection of *V. parahaemolyticus*, facilitating strain typing and strain identification. Molecular biology methods and immunological approaches exhibit high sensitivity and specificity, yet they still require expensive laboratory equipment and intricate operational procedures (11). In comparison, rapid colorimetric detection techniques offer advantages such as simplicity of operation, cost-effectiveness, and intuitive result interpretation (12, 13). *V. parahaemolyticus* detection holds significant implications in terms of food safety and public health. Early detection and control of *V. parahaemolyticus* infection can reduce disease transmission and occurrence, safeguarding food safety and public well-being (14). Therefore, strengthening research on *V. parahaemolyticus* detection, improving detection methods and technologies, and enhancing detection accuracy and sensitivity were of paramount importance (15). The research background in bacterial detection focuses on the utilization of nanoprobes, which are engineered particles with high sensitivity and specificity (16). These nanoprobes have the potential to accurately identify and analyze the presence of bacteria (17), thereby enhancing the efficiency and precision of the bacterial detection method (18). Dual-substrate colorimetry is a commonly used analytical method for quantitatively measuring the concentration of specific substances in a solution (19, 20). This method is based on the chemical reaction between a substrate and the target substance, resulting in observable color changes. The research background also involves the field of nanotechnology. In recent years, the development of nanotechnology has provided new opportunities for dual-substrate colorimetry (2). By harnessing the unique properties of nanomaterials, such as nanoenzyme activity, the sensitivity and selectivity of dual-substrate colorimetry can be enhanced (21). Therefore, researchers are exploring the application of nanomaterials in dual-substrate colorimetry to further improve its performance and expand its application range. Dual-substrate colorimetry, as a widely used analytical technique (22), holds great potential in fields such as analytical chemistry, biomedical sciences, and environmental monitoring (23, 24). Researchers are continuously exploring new methods and materials to enhance the performance and application scope of dual-substrate colorimetry, providing better support for scientific research and practical applications.

In this study, a $Fe_3O_4$-Zn-Mn-Apt probe was developed through the preparation of a $Fe_3O_4$-Zn-Mn conjugated *V. parahaemolyticus* aptamer with oxidase activity, enabling rapid colorimetric detection of *Vibrio parahaemolyticus*. As depicted in Fig. 1, the pre-prepared $Fe_3O_4$-Zn-Mn-Apt was initially subjected to a rotational reaction with the *V. parahaemolyticus* bacterial solution for a duration of 30 min. Subsequently, magnetic separation was performed to eliminate the supernatant, followed by the addition of ortho-phenylenediamine (OPD) and 3,3′,5,5′-tetramethylbenzidine (TMB) solutions for colorimetric reaction. Based on the outcome analysis, varying concentrations of the *V. parahaemolyticus* bacterial solution exhibited diverse degrees of inhibition on the enzymatic activity of $Fe_3O_4$-Zn-Mn. Experimental findings revealed that the nanozyme oxidized OPD initially, and OPD oxidizes TMB. Consequently, under different concentrations of the *VP* bacterial solution, a multitude of colors would manifest, thereby

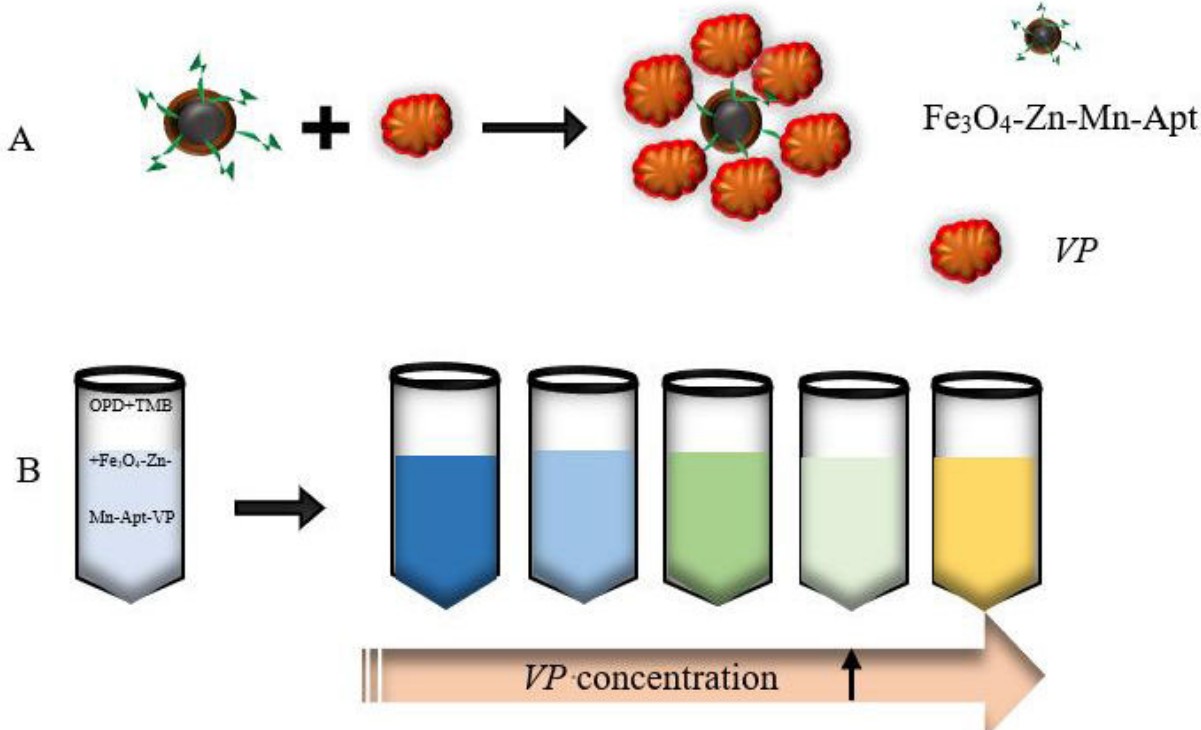

**FIG 1** Schematic diagram of the proposed colorimetric assay for the detection of *V. parahaemolyticus*. (A) The incubation process; (B) the procedures of colorimetry method to detect *V. parahaemolyticus*.

accomplishing a multicolor spectrophotometric detection of *V. parahaemolyticus*. In summary, rapid colorimetric detection techniques offer advantages such as simplicity of operation, cost-effectiveness, and intuitive result interpretation. The development of a $Fe_3O_4$-Zn-Mn-Apt probe with oxidase activity and the rapid colorimetric detection of *V. parahaemolyticus* can be realized. This method holds promising potential for widespread application in areas such as food safety monitoring and medical diagnostics.

## RESULTS AND DISCUSSION

### Assay principle of the developed *VP* colorimetric detection sensor

The feasibility of this approach was validated through several comparative experiments. As depicted in Fig. 2, the catalytic oxidation of OPD to oxOPD by $Fe_3O_4$-Zn-Mn-Apt exhibited a broad absorption band within the range of 350–600 nm, affirming its consistent oxidative effect on OPD. Additionally, $Fe_3O_4$-Zn-Mn-Apt also catalyzed the oxidation of TMB, as evidenced by the presence of UV-vis absorption peaks at 370 and 652 nm for $TMB^+$. When both OPD and TMB were employed as substrates, absorption peaks for oxOPD and $TMB^+$ were observed. Furthermore, an increase in absorbance for oxOPD and a decrease in absorbance for $TMB^+$ were observed, indicating that $TMB^+$ might undergo further oxidation by OPD, generating more oxOPD, while simultaneously being reduced back to TMB. Hence, it can be inferred that this is the key mechanism underlying the generation of the multicolor chromogenic response. In the presence of *V. parahaemolyticus* bacterial solution, a significant decrease in the absorption peak at 650 nm was observed, accompanied by distinct color changes. Therefore, this multicolor chromogenic system holds promise for the detection of *V. parahaemolyticus*.

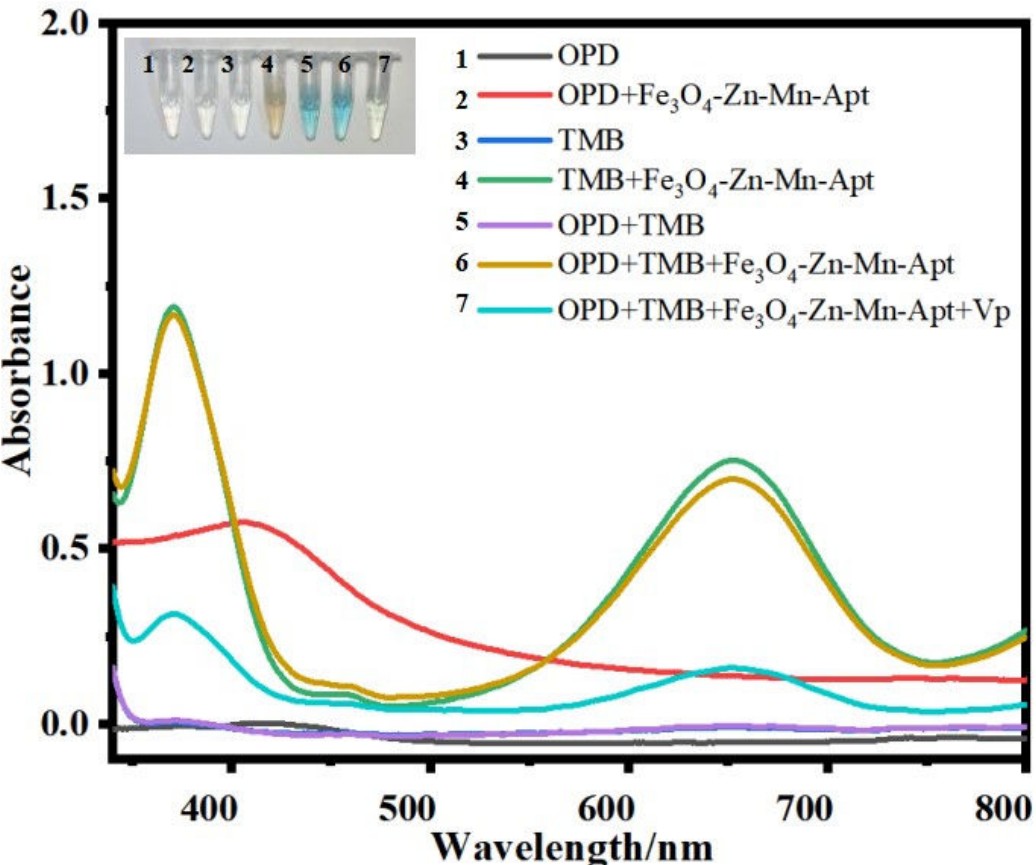

**FIG 2** Feasibility of the developed method.

## Characteristics

We performed a comprehensive characterization of the synthesized magnetic nano-zyme $Fe_3O_4$-Zn-Mn, encompassing a series of systematic investigations. As depicted in Fig. 3A, the synthesized $Fe_3O_4$ nanoparticles exhibited a uniform and well-dispersed spherical porous structure, with an average diameter of 155 nm, consistent with previous reports. Prior to the synthesis of $Fe_3O_4$-Zn-Mn, a series of characterizations were performed. Figure 3B reveals a similar morphology for the $MnO_2$ nanoparticles, with numerous Zn nanoparticles uniformly distributed on the surface of the membrane. Furthermore, elemental mapping and analysis using transmission electron microscopy (TEM)–energy-dispersive X-ray spectroscopy (EDS) confirmed strong signals of Zn and Mn (Fig. S1). It is evident that the Zn signal originates from the Zn nanoparticles, while the Mn signal arises from the $MnO_2$ nanoparticles. Upon coating the Zn-$MnO_2$ onto the surface of $Fe_3O_4$ (Fig. 3C), the average size increased to 195 nm. EDS analysis was employed to determine the elemental composition of the prepared $Fe_3O_4$-Zn-Mn composite material (Fig. 4). The TEM elemental mapping of $Fe_3O_4$-Zn-Mn demonstrated the uniform distribution of Fe, Zn, and Mn throughout the spherical structure, indicating the presence of Zn-Mn on the encapsulated $Fe_3O_4$ surface. Additionally, the elemental analysis results (Fig. S2) revealed simultaneous peaks for Fe, Zn, and Mn elements. These findings confirm the successful integration of Zn-$MnO_2$ with $Fe_3O_4$. Subsequently, TEM imaging of the $Fe_3O_4$-Zn-Mn-Apt probe for *V. parahaemolyticus* captured (Fig. 3D) clearly demonstrates the successful capture of *V. parahaemolyticus* by the $Fe_3O_4$-Zn-Mn-Apt probe. The dual-substrate chromogenic reaction was further analyzed using Fourier-transform infrared (FTIR) spectroscopy (Fig. S3). The obtained spectrum revealed the presence of an -$NH_2$ functional group at 3,454 $cm^{-1}$, attributed to the stretching

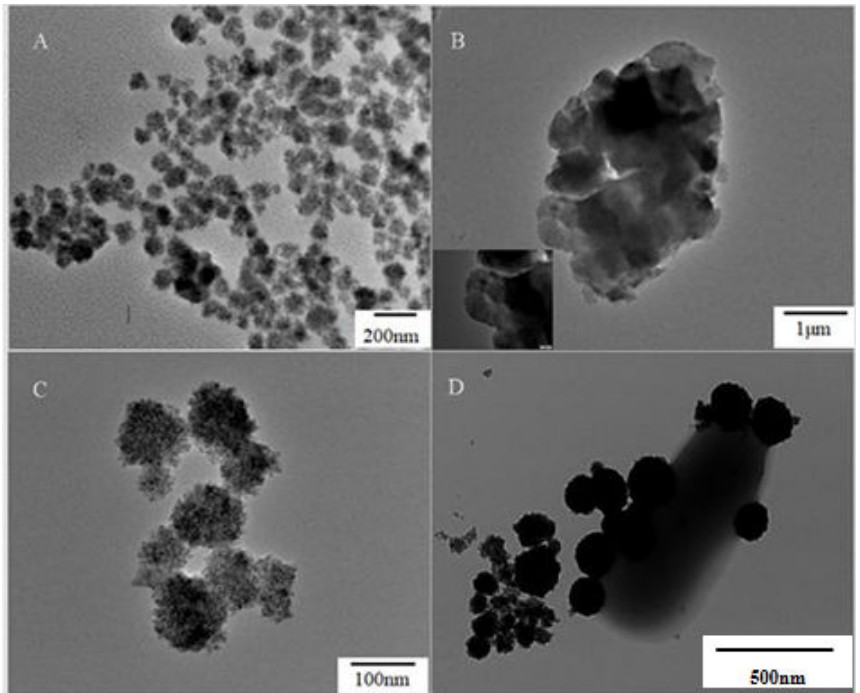

**FIG 3** TEM characterization images: $Fe_3O_4$ (A), $Zn-MnO_2$ (B), $Fe_3O_4$-Zn-Mn (C), and $Fe_3O_4$-Zn-Mn-Apt-*V. parahaemolyticus* (D).

vibration of O-H. Additionally, peaks at 2,358 $cm^{-1}$ indicated the presence of -C≡N, while the bending vibrations of C-O were observed at 1,484 $cm^{-1}$. The presence of -C=C- bonds was confirmed by the peak at 1,634 $cm^{-1}$, and the -C-C- bonds were identified by the peak at 1,083 $cm^{-1}$ (19, 20). These spectral features are likely responsible for the multicolor reaction observed in the dual-substrate system.

## Optimizing conditions for detecting *V. parahaemolyticus*

In order to optimize the performance of the constructed *V. parahaemolyticus* detection system, several factors that could potentially affect the experimental results were investigated. Firstly, the optimal capture performance of the $Fe_3O_4$-Zn-Mn-Apt probe toward *V. parahaemolyticus* was optimized, as shown in Fig. S4. With an increase in the concentration of $Fe_3O_4$-Zn-Mn-Apt, the quantity of *V. parahaemolyticus* in the supernatant gradually decreased, while the capture efficiency sharply increased. When the concentration of $Fe_3O_4$-Zn-Mn-Apt reached 0.5 mg $mL^{-1}$, the capture efficiency reached a remarkable 94.85%, indicating that the majority of *V. parahaemolyticus* in the system could be captured. Therefore, a concentration of 0.5 mg $mL^{-1}$ of $Fe_3O_4$-Zn-Mn-Apt was chosen for subsequent experiments.

For the $Fe_3O_4$-Zn-Mn-Apt probe concentration of 0.5 mg $mL^{-1}$, the influence of TMB concentration on the catalytic effect of $Fe_3O_4$-Zn-Mn was studied, as depicted in Fig. S5A. The absorbance at 652 nm gradually increased with an increase in TMB concentration, but the increasing trend became less pronounced after reaching 8 mM. Subsequently, to construct a multicolor chromogenic system, the impact of OPD on the detection performance was investigated. The detection of *V. parahaemolyticus* was found to be sensitive while ensuring multicolor changes in $Fe_3O_4$-Zn-Mn solutions of different concentrations. As shown in Fig. S5B, the solution stabilized at approximately 0.15 mM OPD. Additionally, Fig. S6 displayed the color changes in the solution under different OPD concentrations (0.05, 0, 0.10, 0.15, 0.20, and 0.25 mM) and different $Fe_3O_4$-Zn-Mn concentrations (0.1, 0.2, 0.3, 0.4, and 0.5 mg). It can be observed that when the OPD concentration was 0.05 mM, the solution only exhibited a bluish green change. As

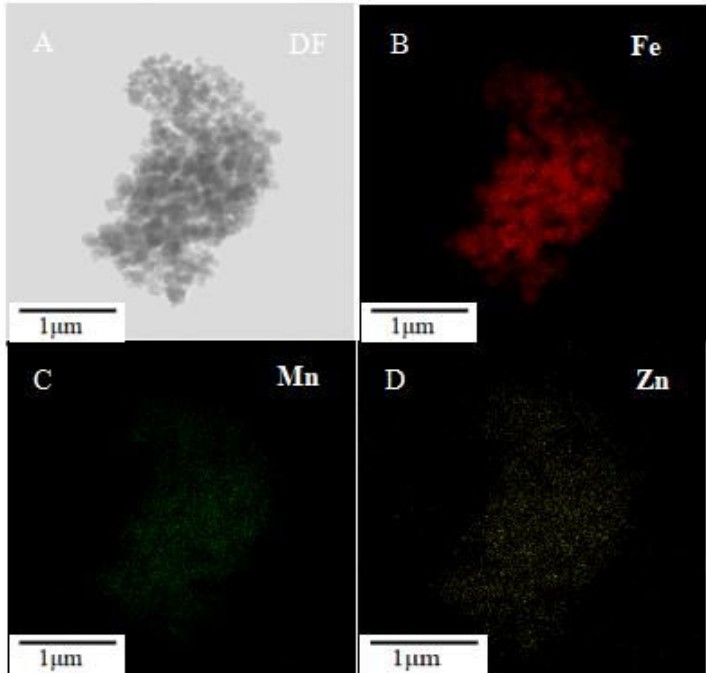

**FIG 4** Elemental mapping and analysis of $Fe_3O_4$-Zn-Mn by TEM energy-dispersive X-ray spectroscopy.

the OPD concentration increased, the proportion of yellow in the solution gradually increased. When the OPD concentration reached 0.10 mM, the solution displayed a single yellow change at different depths. Based on the color contrast chart, under the conditions of 0.10, 0.15, and 0.20 mM OPD, the solution was likely to exhibit multicolor changes. Therefore, considering the results in this figure, an OPD concentration of 0.15 mM was selected to meet the sensitivity requirements for *V. parahaemolyticus* detection.

To achieve the maximum benefit of the color reaction, the same amount of TMB was subjected to different reaction times with $Fe_3O_4$-Zn-Mn catalysis. It was found that the absorbance increased continuously with time and reached a plateau at 5 min, as shown in Fig. S5C. Finally, $Fe_3O_4$-Zn-Mn-Apt was incubated with the same *V. parahaemolyticus* solution for different durations, followed by the addition of the same amount of TMB for color development. The optimal incubation time was determined to be 30 min, as depicted in Fig. S5D. In summary, based on a comprehensive analysis of the experimental results, subsequent experiments were conducted using $Fe_3O_4$-Zn-Mn-Apt (0.5 mg), TMB (8 mM), OPD (0.15 mM), a color development time of 5 min, and an incubation time of 30 min.

## Polycolorimetry for the detection of *V. parahaemolyticus*

Based on the dual-substrate colorimetric method, a variety of colors are generated in the presence of $Fe_3O_4$-Zn-Mn, catalyzed to produce multicolor. Exploiting this phenomenon, a convenient multicolor comparison method has been established for quantitative and semi-quantitative detection of *V. parahaemolyticus*. When *V. parahaemolyticus* is present, certain sites on the surface of $Fe_3O_4$-Zn-Mn-aptamer are occupied, resulting in a gradual decrease in the production of $TMB^{2+}$ with increasing *V. parahaemolyticus* concentrations. Consequently, the solution exhibits a multicolor optical signal. These colors correspond to different concentrations of *V. parahaemolyticus* and can be visually distinguished. As depicted in Fig. 5A, in the absence of *V. parahaemolyticus*, all the $Fe_3O_4$-Zn-Mn-aptamer surfaces catalyze TMB to generate $TMB^+$, displaying a bluish green color. As the concentration of *V. parahaemolyticus* increases, the absorbance at 652 nm gradually decreases. Additionally, the color of the corresponding solution transitions

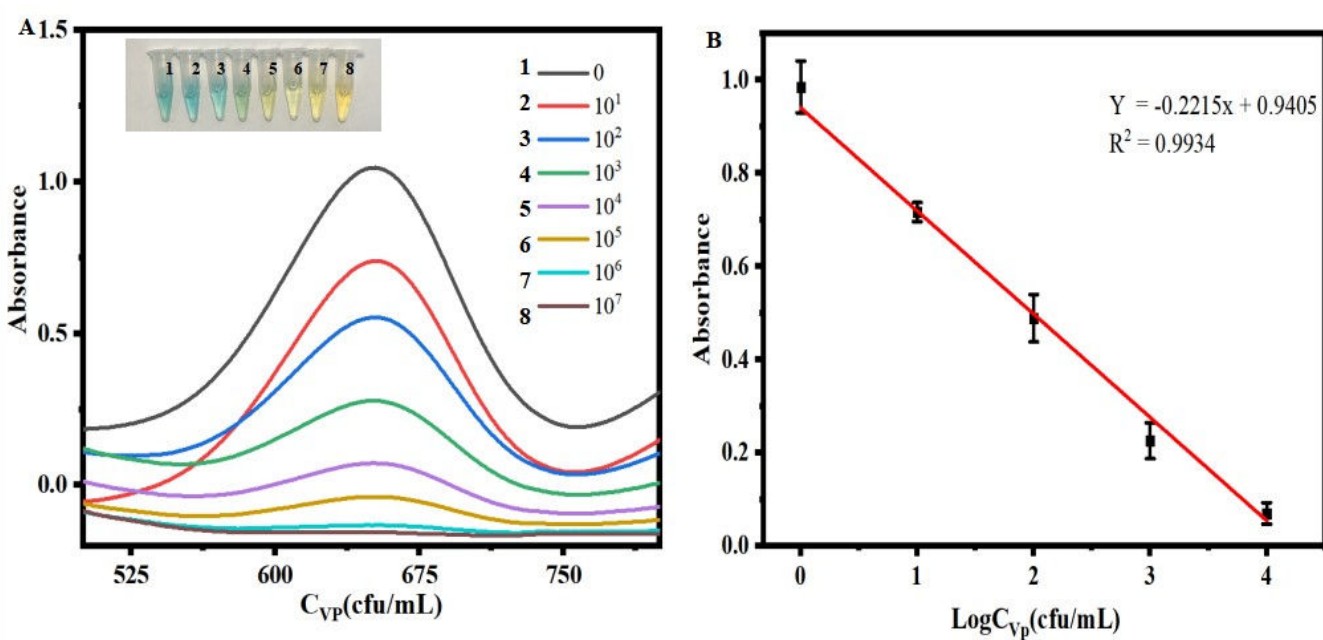

**FIG 5** (A) The UV-vis spectrum corresponding to the colorimetric detection of *Vibrio parahaemolyticus* is presented, accompanied by an illustrative photograph showcasing the corresponding array of colors. (B) Construction of a standard curve using absorbance values at 652 nm and logarithm of TMB concentration ($LogC_{V.\ parahaemolyticus}$).

from bluish green to green and then to yellow, enabling semi-quantitative assessment by the naked eye. The visual detection limit for *V. parahaemolyticus* is 10 cfu mL$^{-1}$, indicating a distinct color difference compared to the control sample. In Fig. 5B, the linear regression equation is $Y = -0.2215X + 0.9405$, with $R^2 = 0.9934$, where $Y$ represents absorbance, $X$ is $LogC_{V.\ parahaemolyticus}$, and $C$ denotes the concentration of *V. parahaemolyticus* (cfu mL$^{-1}$). The limit of detection (LOD) for this method is 1.12 cfu mL$^{-1}$ (LOD = 3σ, where σ is the standard deviation of the blank sample). The multicolor comparison method proposed in this study exhibits outstanding performance in terms of detection time, steps, and limits compared to other reported methods (Table S2).

Therefore, these results demonstrate the feasibility of utilizing the dual-substrate multicolor system for *V. parahaemolyticus* analysis. The method developed in this study possesses a wider linear range and lower LOD detection advantage compared to other methods. Furthermore, the establishment of the dual-substrate colorimetric system endows it with the capability of multicolor comparison. In summary, the aforementioned experimental results highlight the exceptional performance of the developed method in terms of detection time, steps, and limits compared to other reported methods.

### Selective research

The selective and interference studies were conducted to develop a detection method for *V. parahaemolyticus*. Four common pathogens, including *Staphylococcus aureus*, *Listeria monocytogenes*, *Salmonella typhimurium,* and *Escherichia coli O157:H7* (each at a concentration of 10$^3$ mg mL$^{-1}$), were chosen as potential coexisting bacteria. Phosphate buffer solution (PBS) samples were used as reagent blanks. Under optimal conditions, the interference bacteria and *V. parahaemolyticus* were separately added to the detection system, and the detection was performed using a UV-vis spectrophotometer. The blank group, the four common bacteria wells, and the mixed bacteria well all appeared bluish green, indicating that the surface sites of Fe$_3$O$_4$-Zn-Mn-aptamer were not captured. In the presence of *V. parahaemolyticus* alone or in combination with other bacteria, the color of the test tube turned light green, and significant changes in absorbance

were observed in samples containing interference bacteria (Fig. 6). Conversely, when *V. parahaemolyticus* was present, the absorbance change in the sample tube was minimal, indicating that the presence of these interference bacteria had little impact on the detection results. These results demonstrate that the method exhibits high specificity for the test tube and can differentiate it from other bacteria. In summary, the above experimental results highlight the high specificity of the developed method for the test tube and its ability to distinguish it from other bacteria.

## Stability research

A stability analysis was conducted to assess the absorbance and color stability of the test samples over a period of 2 hours. The samples were monitored at regular intervals to evaluate any changes in absorbance and color. As Fig. S7 shows, during the stability analysis, it was observed that the absorbance values remained consistent and did not show significant fluctuations over the 2-hour period. This indicates that the test system maintained its optical properties and provided reliable measurements throughout the duration of the experiment. Furthermore, the color of the samples remained stable and did not exhibit any noticeable changes during the stability analysis. This suggests that the dual-substrate to detect *V. parahaemolyticus* retained its ability to generate the desired color response, ensuring the accuracy and reliability of the detection method. Overall, the stability analysis demonstrated that the developed method exhibits

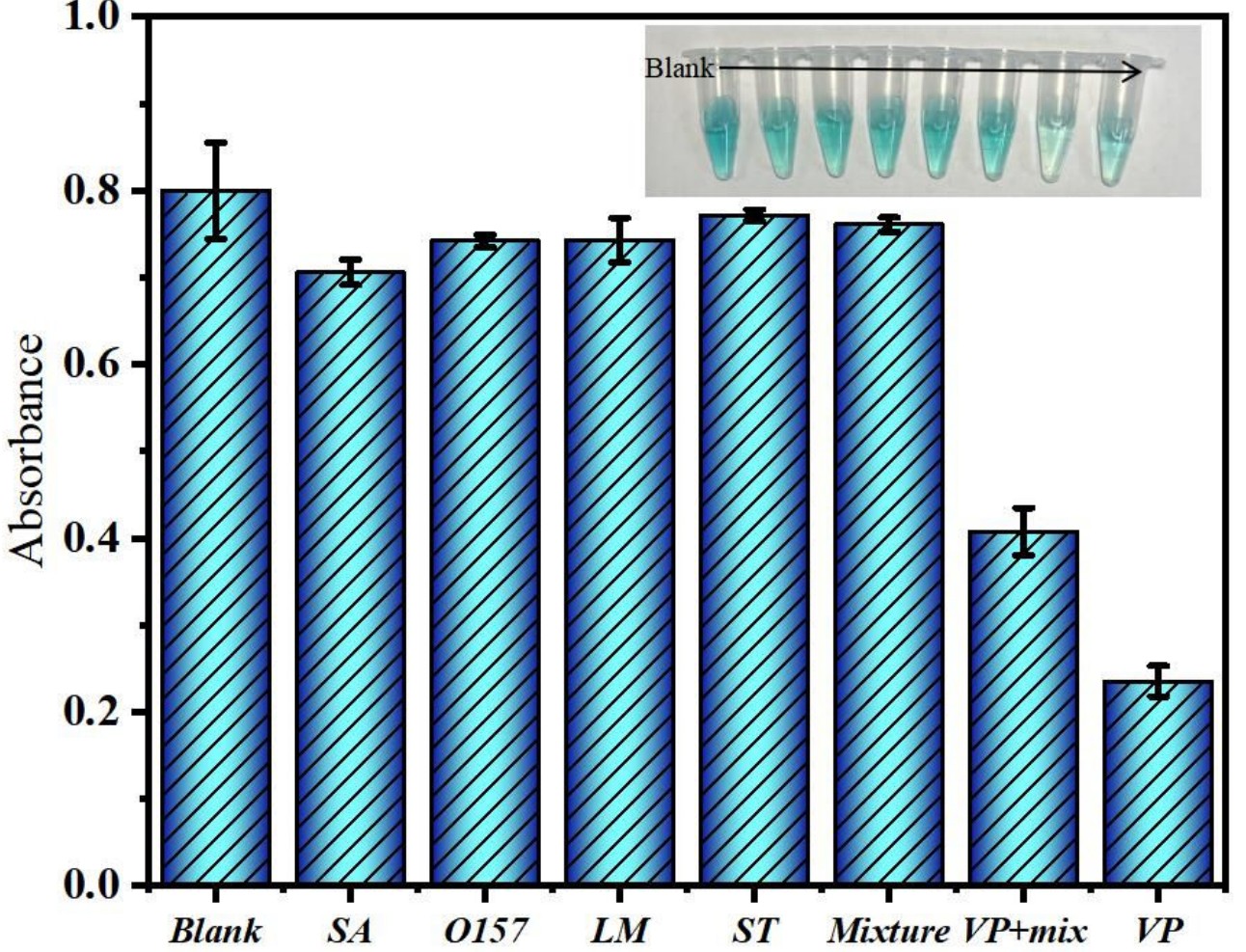

**FIG 6** Selectivity of *V. parahaemolyticus* detection system.

excellent stability in terms of absorbance and color over a 2-hour period. This stability is crucial for obtaining reliable and consistent results in practical applications.

## Real sample analysis

To further investigate the potential application of the developed multicolor method in *V. parahaemolyticus* food analysis, common food samples (mackerel and seaweed) contaminated with *V. parahaemolyticus* were tested using the colorimetric assay to determine the practicality of the method. The food samples were processed according to the literature and tested following the same procedure. Additionally, *V. parahaemolyticus* at 7 and 70 cfu $mL^{-1}$ levels were measured in triplicate in the samples. The summarized results are presented in Table 1, with recovery rates within the range 89.14%–115.86% and RSD within the range 3.4%–13.89% ($n = 3$). These results indicate the potential application of the method for *V. parahaemolyticus* detection in real food samples. In conclusion, the multicolor method shows promise for the analysis of *V. parahaemolyticus*. By successfully detecting and quantifying *V. parahaemolyticus* in common food samples, the method demonstrates its practicality and potential for real-world applications.

## Conclusions

This study presents a sensitive, multicolor, and selective spectrophotometric method for the detection of *V. parahaemolyticus* based on a dual-substrate colorimetric system. The method enables high sensitivity and selective detection of *V. parahaemolyticus*, exhibiting three-color changes in the presence of different concentrations of the bacteria. This allows for both quantitative detection using UV-vis spectrophotometry and semi-quantitative detection by visual observation. The experimental results demonstrate that this sensing method offers a wide linear range, low detection limit, high selectivity, and excellent stability for *V. parahaemolyticus* detection. Notably, the sensor shows great potential for detecting *V. parahaemolyticus* in food samples. This research provides a novel reference for multicolor detection of *V. parahaemolyticus* and opens up possibilities for future exploration of simultaneous multitarget detection in clinical diagnostics, environmental chemistry, and food safety.

## MATERIALS AND METHODS

### Reagents and apparatus

Anhydrous manganese chloride ($MnCl_2$) and *O*-phenylenediamine were purchased from Aladdin Reagents Co., Ltd. (Shanghai, China). Sodium hydroxide (NaOH) and zinc acetate (ZnAc) were acquired from the National Pharmaceutical Group Chemical Reagents Co., Ltd. (Shanghai, China). Bovine serum albumin (BSA) Fraction V and 3,3′,5,5′-tetramethyl-benzidine were received from Shanghai Macklin Biochemical Co., Ltd. (Shanghai, China). N-(3-(Dimethylamino)-propyl)-N′-ethylcarbodiimide hydrochloride (EDC) and N-hydroxy-succinimide (NHS) were purchased from Sigma-Aldrich (USA). All other chemicals and reagents used were of analytical grade and were used without further purification.

All the UV-vis spectra were obtained using the SHIMADZU UV-2600i UV-vis spectro-photometer (SHIMADZU, Japan). Fourier-transform infrared spectroscopy from 4,000 to 500 $cm^{-1}$ was recorded using a Nicolet 6700 FTIR spectrometer (Thermo Inc., USA). All the solutions were prepared through sonication in the ultrasonic cleaner (Kunshan Ultrasonic Instrument Co., Ltd., Kunshan, China).

**TABLE 1** The recoveries and RSD values of detecting bacteria in spiked samples ($\bar{x} \pm s$, $n = 3$)

| Sample | Added (cfu $mL^{-1}$) | Calculated (cfu $mL^{-1}$) | Recovery rate (%) | RSD (%) |
|---|---|---|---|---|
| Mackerel | 7.00 | $7.58 \pm 0.98$ | 108.28 | 12.93 |
| | 70.00 | $81.10 \pm 2.76$ | 115.86 | 3.40 |
| Seaweed | 7.00 | $6.24 \pm 0.36$ | 89.14 | 5.78 |
| | 70.00 | $77.52 \pm 8.68$ | 110.74 | 11.20 |

## Preparation of nanomaterials

The magnetic nanoparticles of $Fe_3O_4$ were prepared using the previous method (20). The Zn-Mn nanomaterial was synthesized through a hydrothermal process. In brief, manganese chloride (100 mM, 2 mL) and BSA (0.05 g) were dissolved in 10 mL of pure water and reacted in a 50-mL beaker for 30 min with vigorous stirring. Sodium hydroxide (70 mM, 10 mL) was added, and the reaction continued for 5 min. Then, a solution of zinc acetate (10 mM, 20 mL) was dropwise-added, and the reaction proceeded for 12 hours. The mixture was centrifuged at 12,000 rpm for 15 min, followed by three washes with anhydrous ethanol to collect the precipitate. The obtained precipitate was dried overnight at 50°C. $Fe_3O_4$ and Zn-Mn were combined in a hydrothermal process at 50°C with intense stirring, resulting in the formation of $Fe_3O_4$-Zn-Mn magnetic nanoenzymes. $Fe_3O_4$-Zn-Mn magnetic nanoenzymes were then coupled with the *V. parahaemolyticus*-Apt (100 µL) through EDC (10 mg/mL) and NHS (10 mg mL$^{-1}$), yielding the $Fe_3O_4$-Zn-Mn-Apt probe.

## Polycolorimetry for the detection of *V. parahaemolyticus*

First, 0.5 mg/mL of $Fe_3O_4$-Zn-Mn-Apt and 1 mL of *V. parahaemolyticus* solution of different concentrations were added to the sample vial and incubated at a mixer for 30 min, followed by magnetic separation and de-supernatant. Next, 500 µL of OPD solution (0.15 mM) and 500 µL of TMB solution (8 mM) were incubated for 5 min at room temperature. After magnetic separation, the supernatant was removed into a 1.5-mL centrifuge tube, and color changes and UV-vis spectra in the range of 500–800 nm were recorded. The *V. parahaemolyticus* concentration and absorbance values (650 nm) were used to construct the standard curve.

## ACKNOWLEDGMENTS

The authors gratefully acknowledge the financial supports by the Chinese National Natural Science Foundation (Grant No. 82003502 and 21904053), the Science and Technology Plan Project of Yantai (Grant No. 2022XDRH004), the Technology Development Commissioned Project Fund (2022LHX110), the Natural Science Foundation of Shandong Province (Grant No. ZR2020QB086), the Youth Innovation Technology Project of Higher School in Shandong Province (Food Nanotechnology Innovation Team), and the Shandong Key Laboratory of Biochemical Analysis, College of Chemistry and Molecular Engineering (SKLBA2307).

## AUTHOR AFFILIATIONS

[1]College of Food Engineering, Ludong University, Yantai, Shandong, China
[2]Bio-Nanotechnology Research Institute, Ludong University, Yantai, Shandong, China
[3]Shandong Key Laboratory of Biochemical Analysis, College of Chemistry and Molecular Engineering, Qingdao University of Science and Technology, Qingdao, China
[4]Yantai Laishan District Center for Disease Control and Prevention, Centers for Disease Control and Prevention, Yantai, Shandong, China

## AUTHOR ORCIDs

Wenteng Qiao http://orcid.org/0009-0005-9014-0000
Yushen Liu http://orcid.org/0009-0007-6974-5982

## FUNDING

| Funder | Grant(s) | Author(s) |
| --- | --- | --- |
| MOST \| National Natural Science Foundation of China (NSFC) | 82003502 | Yushen Liu |

| Funder | Grant(s) | Author(s) |
|---|---|---|
| MOST \| National Natural Science Foundation of China (NSFC) | 21904053 | Luliang Wang |
| The science and technology plan project of Yantai | 2022XDRH004 | Yushen Liu |
| Shandong Key Laboratory of Biochemical Analysis (SKLBA) | SKLBA2307 | Yushen Liu |

## AUTHOR CONTRIBUTIONS

Luliang Wang, Funding acquisition, Project administration, Resources | Kun Yang, Formal analysis | Yushen Liu, Data curation, Funding acquisition, Writing – review and editing | Quanwen Liu, Investigation, Writing – review and editing | Feng Yin, Investigation, Writing – review and editing.

## ADDITIONAL FILES

The following material is available online.

### Supplemental Material

**Supplemental material (Spectrum03189-23-s0001.docx).** Fig. S1 to S7; Tables S1 and S2.

### Open Peer Review

**PEER REVIEW HISTORY (review-history.pdf).** An accounting of the reviewer comments and feedback.

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
