## [Reviewer comments · Microbiology Spectrum]

Microbiology Spectrum

A multichromatic colorimetric detection method for *Vibrio parahaemolyticus* based on Fe₃O₄-Zn-Mn nanoenzyme and dual substrates

qiao wenteng, Luliang Wang, Kun Yang, Yushen Liu, Quanwen Liu, and Feng Yin

Corresponding Author(s): qiao wenteng, Ludong University Library

Review Timeline:

Submission Date:	August 28, 2023
Editorial Decision:	September 23, 2023
Revision Received:	September 28, 2023
Editorial Decision:	October 25, 2023
Revision Received:	October 26, 2023
Accepted:	November 6, 2023

Editor: Sadjia Bekal

Reviewer(s): Disclosure of reviewer identity is with reference to reviewer comments included in decision letter(s). The following individuals involved in review of your submission have agreed to reveal their identity: Kun Xu (Reviewer #1)

Transaction Report:

DOI: <https://doi.org/10.1128/spectrum.03189-23>

September 21, 2023

Mr. qiao wenteng
Ludong University Library
hongqizhonglu
Yantai
China

Re: Spectrum03189-23 (**A multichromatic colorimetric detection method for *Vibrio parahaemolyticus* based on Fe₃O₄-Zn-Mn nanoenzyme and dual substrates**)

Dear Mr. qiao wenteng:

Link Not Available

Sincerely,

Sadjia Bekal

Journals Department
Reviewer comments:

Reviewer #1 (Comments for the Author):

This manuscript introduced a new a dual-substrate colorimetric system for rapid and multi- colorimetric semi-quantification of *V. parahaemolyticus*. The present study is fairly complete in its breadth and the results are decent. However, several technical and formal flaws so that a minor revision is mandatory.

- 1, The manuscript contains inconsistencies in the abbreviations for the more critical nanomases, such as Fe₃O₄-Zn-Mn, Fe₃O₄-Zn-MnO₂, Fe-Zn-Mn.
- 2, The units of the manuscript and Electronic Supplementary Material do not match, please unify and confirm.
- 3, Some content need to be modified.

- (1)Line 113 "UV absorption peaks", Please correct the format.
- (2)Line 140 "Figure 2D", Please be consistent with the figure.
- (3)Line 217 and 300 "mg/mL", Please write properly.
- (4) The bacterial abbreviations in Fig 5 of the manuscript and the Electronic Supplementary Material are inconsistent, please correct.

Reviewer #2 (Comments for the Author):

This manuscript reports a novel nanocomposite catalytic bisubstrate system for rapid and multi-colorimetric semi-quantitative detection of *Vibrio parahaemolyticus*. The topic is interesting, and the results are sound. I recommend minor revisions as noted below:

- (1) The color writing in Figure 5 is inconsistent.
- (2) The abbreviations for a few critical substances are inconsistent, such as nanocomposites, bacteria.
- (3) The units used in the manuscript and the electronic supplement are inconsistent.
- (4) Missing scale bar in Figure 2D.
- (5) The unit symbols are not standardized.

Staff Comments:

Preparing Revision Guidelines

Please return the manuscript within 60 days; if you cannot complete the modification within this time period, please contact me. If you do not wish to modify the manuscript and prefer to submit it to another journal, please notify me of your decision immediately so that the manuscript may be formally withdrawn from consideration by Microbiology Spectrum.

This manuscript introduced a new a dual-substrate colorimetric system for rapid and multi-colorimetric semi-quantification of *V. parahaemolyticus*. The present study is fairly complete in its breadth and the results are decent. However, several technical and formal flaws so that a minor revision is mandatory. The present manuscript is suggested for publication after issues listed below are addressed:

- 1, The manuscript contains inconsistencies in the abbreviations for the more critical nanomases, such as Fe₃O₄-Zn-Mn, Fe₃O₄-Zn-MnO₂, Fe-Zn-Mn.
- 2, The units of the manuscript and Electronic Supplementary Material do not match, please unify and confirm.
- 3, Some content need to be modified.
 - (1) Line 113 “UV absorption peaks”, Please correct the format.
 - (2) Line 140 “Figure 2D”, Please be consistent with the figure.
 - (3) Line 217 and 300 “mg/mL”, Please write properly.
 - (4) The bacterial abbreviations in Fig 5 of the manuscript and the Electronic Supplementary Material are inconsistent, please correct.

This manuscript reports a novel nanocomposite catalytic bisubstrate system for rapid and multi-colorimetric semi-quantitative detection of *Vibrio parahaemolyticus*. The topic is interesting, and the results are sound. I recommend minor revisions as noted below:

- (1) The color writing in Figure 5 is inconsistent.
- (2) The abbreviations for a few critical substances are inconsistent, such as nanocomposites, bacteria.
- (3) The units used in the manuscript and the electronic supplement are inconsistent.
- (4) Missing scale bar in Figure 2D.
- (5) The unit symbols are not standardized.

Response to reviews

Dear editor and reviews

Thank you very much for your review and valuable comments. We sincerely accept these comments and suggestions, and made the corresponding changes. In the revision, two new authors have given great help, giving sincere suggestions and useful methods. Make a greater contribution to the revision of the manuscript. We ask that they be listed as co-authors. All changes made to the text are highlighted in yellow for easy identification. The following are the main corrections and responses to the reviewers' comments:

Response to comments of Reviewer 1 :

1, The manuscript contains inconsistencies in the abbreviations for the more critical nanomesas, such as $\text{Fe}_3\text{O}_4\text{-Zn-Mn}$, $\text{Fe}_3\text{O}_4\text{-Zn-MnO}_2$, Fe-Zn-Mn .

Response: Thank you for your precise review. We apologize for the careless writing error. We have revised the manuscript in lines 18,35,125-126,440,443 and support information in lines 20, 38 as $\text{Fe}_3\text{O}_4\text{-Zn-Mn}$.

2, The units of the manuscript and Electronic Supplementary Material do not match, please unify and confirm.

Response: Thank you for your careful checks. We apologize for the careless writing error. We have unified the manuscript with the supporting literature.

3, Some content needs to be modified.

(1) Line 113 "UV absorption peaks", Please correct the format.

(2) Line 140 "Figure 2D", Please be consistent with the figure.

(3) Line 217 and 300 "mg/mL", Please write properly.

(4) The bacterial abbreviations in Fig 5 of the manuscript and the Electronic Supplementary Material are inconsistent, please correct.

Response: We were really sorry for our careless mistakes. Thank you for your reminder. We have made the following corrections:

(1) We have revised the manuscript in lines 114 as UV-vis absorption peaks.

(2) We have revised the manuscript in lines 138 as Fig 2D.

(3) We have revised the manuscript in lines 218,298 as mg mL^{-1} .

(4) We have revised the manuscript in lines and support information as *V. parahaemolyticus* to make the word harmonized within the whole manuscript.

Response to comments of Reviewer 2 :

1, The color writing in Figure 5 is inconsistent.

Response: We sincerely thank the reviewer for careful reading. As suggested by the reviewer. We have revised the manuscript in lines 221 as light green.

2, The abbreviations for a few critical substances are inconsistent, such as nanocomposites, bacteria.

Response: Thank you for your valuable recommendations. We apologize for the careless writing error. We have revised some place of the manuscript and support information as Fe₃O₄-Zn-Mn and *V. parahaemolyticus*.

3, The units used in the manuscript and the electronic supplement are inconsistent.

Response: Thank you for your valuable recommendations. We apologize for the careless writing error. We have revised the manuscript in lines 204 as cfu mL⁻¹.

4, Missing scale bar in Figure 2D.

Response: We think this is an excellent suggestion. We apologize for the careless writing error. We have re-written this part according to Reviewer's suggestion.

5, The unit symbols are not standardized.

Response: Thank you for your valuable recommendations. We apologize for the careless writing error. We have revised some unit symbols as cfu mL⁻¹ and mg mL⁻¹.

Re: Spectrum03189-23R1 (**A multichromatic colorimetric detection method for *Vibrio parahaemolyticus* based on Fe₃O₄-Zn-Mn nanoenzyme and dual substrates**)

Dear Mr. qiao wenteng:

We noticed that new authors have been added on the revised version. Please, send us a document signed by all authors accepting this modification.

Sincerely,
Sadjia Bekal
Editor
Microbiology Spectrum

Reviewer #2 (Comments for the Author):

The authors have properly addressed my previous comments.